# A Combinatorial Semi-Bandit Approach to Charging Station Selection for Electric Vehicles

**Niklas Åkerblom**                                                            *niklas.akerblom@volvocars.com*
*Volvo Car Corporation*
*Chalmers University of Technology*

**Morteza Haghir Chehreghani**                                                 *morteza.chehreghani@chalmers.se*
*Chalmers University of Technology*

**Reviewed on OpenReview:** *https://openreview.net/forum?id=ndw9OpkNM9*

## Abstract

In this work, we address the problem of long-distance navigation for battery electric vehicles (BEVs), where one or more charging sessions are required to reach the intended destination. We consider the availability and performance of the charging stations to be unknown and stochastic, and develop a combinatorial semi-bandit framework for exploring the road network to learn the parameters of the queue time and charging power distributions. Within this framework, we first outline a method for transforming the road network graph into a graph of feasible paths between charging stations to handle the constrained combinatorial optimization problem in an efficient way. Then, for the feasibility graph, we use a Bayesian approach to model the stochastic edge weights, utilizing conjugate priors for the one-parameter exponential and two-parameter gamma distributions, the latter of which is novel to multi-armed bandit literature. Finally, we apply combinatorial versions of Thompson Sampling, BayesUCB and Epsilon-greedy to the problem. We demonstrate the performance of our framework on long-distance navigation problem instances in large-scale country-sized road networks, with simulation experiments in Norway, Sweden and Finland[1].

## 1 Introduction

In the coming years, it will be crucial for society to shift the transport sector (personal and commercial) towards electrification, to reach global targets on reduced greenhouse gas emissions. *Range anxiety* is still a major obstacle to the widespread adoption of battery electric vehicles (BEVs) for transportation. This phenomenon can be characterized as the fear that (potential or current) BEV drivers might feel about exceeding the electric range of their vehicle before reaching either their destination or a charging station, thus being stranded with an empty battery.

There is another, perhaps less commonly discussed, but potentially more relevant issue called *charging anxiety*. Whereas the range of typical electric vehicles, while still shorter than that of combustion engine vehicles, has been steadily increasing, the act of charging the battery is still, often, a cumbersome task. Even at locations with fast chargers, various factors may severely impact the total travel time of a particular trip. At the highest possible charging power, charging a BEV battery from nearly empty to close to maximum capacity may take more than 30 minutes. However, maximum power might not always be provided, in practice. Furthermore, queues to charging stations may appear due to the (relatively) long charging times and few charging locations. Issues like these might diminish public trust in BEVs as viable alternatives to combustion engine vehicles.

---

[1]Our code is available at `https://github.com/volvo-cars/eene-nav-bandit-sim`

In this work, we attempt to mitigate any charging anxiety arising due to the aforementioned factors by developing an online self-learning algorithmic framework for navigation of BEVs, capable of taking such charging issues into account. We view the task as a *sequential decision-making problem under uncertainty* and model it as a *combinatorial semi-bandit problem* to address the trade-off between exploring charging stations to learn more information about them and exploiting previously collected knowledge to select charging stations that are likely to be good. Within this framework, we employ a Bayesian approach, with conjugate priors novel to bandit literature. To our knowledge, our work is the first study that addresses the challenging real-world problem of charging station selection in the partial information setting using multi-armed bandit (MAB) methods. Thereby, our work provides a novel framework to develop and investigate advanced multi-armed (combinatorial) bandit methods. As mentioned, our work is also one of the few large-scale real-world applications of multi-armed bandits, specifically in the combinatorial setting. To achieve such scalability, we transform the road graph into a feasibility graph, where feasible paths between charging stations are pre-computed to improve run-time efficiency. Such a transformation of the problem instances for the purpose of computational efficiency is novel to the MAB community.

## 2 Related work

Several works have studied shortest path algorithms to address the problem of energy efficient navigation, e.g., Artmeier et al. (2010); Sachenbacher et al. (2011), focusing on minimizing energy consumption, and Baum et al. (2015), studying how to minimize travel time while ensuring that battery energy is not fully depleted (utilizing charging stations, if necessary). Sweda et al. (2017) outline a Dynamic Programming (DP) approach for adaptive routing of electric vehicles, and model charging station availability and queue / waiting times, but assume that all distributions are known in advance. A similar work by Guillet et al. (2022) formulates charging station selection as a stochastic search problem, addressed with a DP-based approach. BEV navigation problems (including charging station selection) have been modelled as detailed reinforcement learning problems (Lee et al., 2020; Qian et al., 2020), but often with high computational demands making them infeasible for long-distance navigation. To our knowledge, this problem has not been studied in the partial information setting using multi-armed bandits, and our work is the first contribution of this kind. Prior works on multi-armed bandits for energy-efficient navigation, e.g., Åkerblom et al. (2020; 2023a), are significantly simplified without considering travel time and charging. Dealing with charging, in particular in a computationally efficient way, requires considerably more sophisticated models and methods, beyond the existing applications of (combinatorial) multi-armed bandit methods, as will be demonstrated in this paper.

In general, the multi-armed bandit problem is a versatile way of describing how to utilize limited resources to balance exploration of an environment to gain new knowledge and usage of previously collected knowledge to increase long-term reward. Thompson Sampling (Thompson, 1933) is an early algorithm attempting to address this trade-off, which has recently increased in popularity due to demonstrated experimental performance (Graepel et al., 2010; Chapelle & Li, 2011) and proven theoretical performance guarantees (Agrawal & Goyal, 2012; Kaufmann et al., 2012b; Bubeck & Liu, 2013; Russo & Van Roy, 2014).

Another type of method commonly used for sequential decision-making problems is the Upper Confidence Bound (UCB) (Auer, 2002) algorithm. Like Thompson Sampling, this method has been adapted to many different settings, including combinatorial optimization problems (Chen et al., 2013). UCB methods have also been used for MAB problems with sub-exponential rewards (Jia et al., 2021), e.g., for selection of bike rental companies with exponential service times, but not in combinatorial settings as far as we are aware.

## 3 Model

In this section, we describe how to represent the road network as a graph and how to transform it into a graph of feasible paths between charging stations to allow for computationally efficient charging station selection. Furthermore, we outline an approach for probabilistic modelling of the queue time (i.e., the time spent waiting for an occupied charging station) and the charging power of each charging station. In Appendix A, we include a summary of the notation used throughout this work.

### 3.1 Road network graph

We model the *underlying* road network using a directed and weighted graph $\mathcal{G}^{\mathrm{road}} \left( \mathcal{V}^{\mathrm{road}}, \mathcal{E}^{\mathrm{road}}, \boldsymbol{\tau}^{\mathrm{road}} \right)$. Each vertex $u \in \mathcal{V}^{\mathrm{road}}$ corresponds to *either* an intersection or some other important location of the road network (e.g., a charging station). Each directed edge $e \in \mathcal{E}^{\mathrm{road}}$ represents a road segment from a location $u \in \mathcal{V}^{\mathrm{road}}$ to another location $u' \in \mathcal{V}^{\mathrm{road}}$, which we may also indicate by writing $e = (u, u')$. We further denote the travel time of each road segment $e \in \mathcal{E}^{\mathrm{road}}$ as $\tau_e^{\mathrm{road}}$, and the vector of all edge travel times as $\boldsymbol{\tau}^{\mathrm{road}}$. Additionally, each road segment $e \in \mathcal{E}^{\mathrm{road}}$ has an associated energy consumption $\varepsilon_e^{\mathrm{road}}$, which is the energy needed by a given vehicle to traverse the complete road segment. A convention we follow throughout the paper is to indicate vectors with bold symbols, with individual elements indexed by edge or vertex subscripts. Furthermore, we let $\mathcal{V}^{\mathrm{charge}} \subseteq \mathcal{V}^{\mathrm{road}}$ be the set of locations which contain charging stations. Each charging location $u \in \mathcal{V}^{\mathrm{charge}}$ is associated with a maximum charging power $\varrho_u^{\mathrm{max}}$ which the charging station is able to provide. We also assume that there is a corresponding value for the minimum charging power $\varrho_u^{\mathrm{min}}$ available at each station. With the exception of the actual charging power $\varrho_u^{\mathrm{charge}} \in \left[ \varrho_u^{\mathrm{min}}, \varrho_u^{\mathrm{max}} \right] \subseteq \mathbb{R}^+$ and the queue time $\tau_u^{\mathrm{queue}} \in \mathbb{R}^+$, both of which we consider stochastic and unknown, all other parameters mentioned in this work (except the charging time, which directly depends on $\varrho_u^{\mathrm{charge}}$) are assumed to be fixed and known by the learning agent.

A connected sequence $\boldsymbol{p}$ of edges (or vertices, equivalently) is called a *path* through the graph. Given a source vertex $u^{\mathrm{src}} \in \mathcal{V}^{\mathrm{road}}$ and a target vertex $u^{\mathrm{trg}} \in \mathcal{V}^{\mathrm{road}}$ (both assumed to be fixed and known), we denote the set of all paths starting in $u^{\mathrm{src}}$ and ending in $u^{\mathrm{trg}}$ as $\mathcal{P}_{(u^{\mathrm{src}}, u^{\mathrm{trg}})}^{\mathrm{road}}$. Assuming that we aim to find a path which minimizes the total travel time, we let, for each edge $e \in \mathcal{E}^{\mathrm{road}}$, the *edge weight* be $\tau_e^{\mathrm{road}}$. Then, the *shortest path problem*, given $u^{\mathrm{src}}$, $u^{\mathrm{trg}}$ and $\mathcal{G}^{\mathrm{road}}$ is defined as

$$\boldsymbol{p}^* = \arg \min_{\boldsymbol{p} \in \mathcal{P}_{(u^{\mathrm{src}}, u^{\mathrm{trg}})}^{\mathrm{road}}} \left( \sum_{e \in \boldsymbol{p}} \tau_e^{\mathrm{road}} \right), \tag{1}$$

which may be addressed by one of several classical methods, e.g., Dijkstra's algorithm (Dijkstra, 1959), the Bellman-Ford algorithm (Shimbel, 1954; Ford Jr, 1956; Bellman, 1958) or the A* algorithm (Hart et al., 1968). The A* algorithm, in particular, can be described as a *best-first search* method (see e.g., Dechter & Pearl 1985), where a provided heuristic function is used to guide the algorithm towards promising solutions. An admissible heuristic function should be able to provide an underestimate of the total weight of any path between a pair of given vertices. For a road network graph with travel time edge weights, such as $\mathcal{G}^{\mathrm{road}}$, we can use a function which calculates a travel time value based on the maximum allowed speed in the road network and the beeline distance between the two vertices. When a good heuristic function is used, the A* algorithm is computationally more efficient than Dijkstra's algorithm, while still guaranteeing that the optimal path is found.

### 3.2 Construction of feasibility graph

The model described in the previous section is sufficient for many applications. Since fossil fuel stations are ubiquitous in most road networks, and since the time required for refueling is typically negligible, the model can be used for navigation of combustion engine vehicles without significant modifications. For BEVs, however, charging can take more than 30 minutes, and multiple charging sessions may be required for longer trips. These factors, combined with the relative sparsity of the charging infrastructure, means that charging should not be disregarded in the navigation problem.

The time spent on charging depends on the amount of energy needed and the charging power provided. Furthermore, queues may occur if all charging stations at a particular location are occupied at the same time. The *resource-constrained shortest path problem* (RCSPP) (Joksch, 1966) is a variant of the shortest path problem where each edge is associated with one or more resources, in addition to the weight. The problem, which is known to be NP-complete (Handler & Zang, 1980) for even a single resource per edge, is to find the shortest path (w.r.t. the weights) such that the accumulated resources for the path do not exceed specified constraints. The single-resource version of RCSPP is a special case of our problem (with consumed energy as the resource and battery capacity as the constraint), where it is also possible to replenish (recharge) the resource at a subset of the vertices, making it necessary to approximate the problem.

In this work, for simplicity, we assume that each charging session has to fully charge the battery. We also assume that the paths between charging stations should be chosen to minimize travel time, even if there are alternative paths with less energy consumption. For clarity, throughout this work when the battery is stated to be either empty or fully charged, the battery state of charge is actually 10% or 80%, respectively, for safety and durability reasons.

With the aforementioned two assumptions, it is possible to transform the road graph into a *feasibility graph*, where feasible paths between charging stations are pre-computed to improve run-time efficiency. We denote this directed and weighted graph $\mathcal{G}^{\text{feasible}}\left(\mathcal{V}^{\text{feasible}}, \mathcal{E}^{\text{feasible}}, \boldsymbol{\tau}^{\text{feasible}}\right)$. In order to construct the feasibility graph, we start by letting $\mathcal{V}^{\text{feasible}} = \mathcal{V}^{\text{charge}}$ be the set of charging stations. Then, for any given path $\boldsymbol{p}$ through $\mathcal{G}^{\text{road}}$, let $\tau_{\boldsymbol{p}}^{\text{road}} = \sum_{e \in \boldsymbol{p}} \tau_e^{\text{road}}$ be the travel time of the path and $\varepsilon_{\boldsymbol{p}}^{\text{road}} = \sum_{e \in \boldsymbol{p}} \varepsilon_e^{\text{road}}$ be the total energy consumption of the path. We create a new set of edges $\mathcal{E}^{\text{path}} = \left\{(u, u') \in \mathcal{V}^{\text{feasible}} \times \mathcal{V}^{\text{feasible}}\right\}$, where each edge $(u, u') \in \mathcal{E}^{\text{path}}$ corresponds to the shortest path $\boldsymbol{p}_{(u,u')}^* = \arg\min_{\boldsymbol{p} \in \mathcal{P}_{(u,u')}^{\text{road}}} \tau_{\boldsymbol{p}}^{\text{road}}$ between the charging stations (computed as in Eq. 1), where we denote $\tau_{(u,u')}^{\text{path}} = \tau_{\boldsymbol{p}_{(u,u')}^*}^{\text{road}}$ and $\varepsilon_{(u,u')}^{\text{path}} = \varepsilon_{\boldsymbol{p}_{(u,u')}^*}^{\text{road}}$. Finally, given a maximum battery capacity $\varepsilon^{\text{max}}$ and minimum battery capacity $\varepsilon^{\text{min}}$, we define the set of feasible edges as the set of shortest paths between charging stations where the battery capacity exceeds the energy consumption, i.e.,

$$\mathcal{E}^{\text{feasible}} = \left\{e \in \mathcal{E}^{\text{path}} \mid \varepsilon_e^{\text{path}} \le \varepsilon^{\text{max}} - \varepsilon^{\text{min}}\right\}. \tag{2}$$

Moreover, we define a vector of travel times for the edges of the feasibility graph, $\boldsymbol{\tau}^{\text{feasible}}$. For each edge $(u, u') \in \mathcal{E}^{\text{feasible}}$, we let

$$\tau_{(u,u')}^{\text{feasible}} = \tau_{(u,u')}^{\text{path}} + \tau_{u'}^{\text{queue}} + \tau_{(u,u')}^{\text{charge}} , \tag{3}$$

where the charging time $\tau_{(u,u')}^{\text{charge}} = \varepsilon_{(u,u')}^{\text{path}} / \varrho_{u'}^{\text{charge}}$ depends on both the energy consumed on the edge $(u, u')$ and the provided charging power at $u'$, while we assume that the queue time $\tau_{u'}^{\text{queue}}$ only depends on $u'$. Note that $\varepsilon_{(u,u')}^{\text{path}} = \tau_{(u,u')}^{\text{charge}} \cdot \varrho_{u'}^{\text{charge}}$ since we assume that the energy we need to charge at $u'$ is the same as the energy consumed while traversing the preceding path $(u, u')$.

We emphasize that even though we assume each charging session fully charges the battery, the construction of the feasibility graph (and the entire online learning framework presented in this work) can be extended in a straightforward way to allow for partial charging. If the battery energy available at the start of each path is discretized into a finite number of *energy levels*, vertices can be added for these to each charging station, where edges between them represent partial charging choices. Then, feasibility graph *layers* may be constructed for each of the energy levels, without increasing the number of unknown parameters that an agent needs to estimate. See, e.g., Sweda et al. (2017), for another approach using discretized charging levels for partial charging, applied to a setting with fixed and known parameters.

### 3.3 Probabilistic queue and charging times

As stated earlier, we consider both the queue time and the charging power of each charging station to be stochastic and unknown, only to be revealed after the station has been visited. The charging power can easily be measured using internal signals of the vehicle, while the queue time can be inferred from the time spent in close proximity to the charging station (e.g., using a GPS sensor or an odometer) before charging. In contrast, we assume that we are given the travel time and energy consumption of each road segment in the road network graph (and, in practice, that they are fixed and known by the agent). We further assume that the queue time and charging power are independently distributed, both with respect to each other, as well as between different charging stations. In reality, they exhibit a complex interdependence, where low charging power might cause queues to appear, and the simultaneous charging of many vehicles may cause the available power to decrease.

#### 3.3.1 Queue time model

The queuing behavior at a particular charging location may be complex, depending on the characteristics of the location. A charging location has few or many charging stations, where each may have multiple

connectors. The stations may also differ in the maximum charging power provided, as well as the price of charging. These, and other factors, impact the preferences of drivers towards different stations, especially if many of the stations at the same location are occupied simultaneously.

Rather than modelling the queues in detail, we take inspiration from a simple model of queuing theory, the *M/M/1 queue* (Kendall, 1953), and assume that the queue time $\tau_u^{\text{queue}}$ of each charging station $u \in \mathcal{V}^{\text{feasible}}$ is exponentially distributed according to an unknown rate parameter $\lambda_u^{\text{queue}}$. The likelihood function of the queue time model can then be defined as

$$P\left(\tau_u^{\text{queue}}|\lambda_u^{\text{queue}}\right) = \text{Exp}(\lambda_u^{\text{queue}}). \tag{4}$$

We also take a Bayesian view and assume that the rate parameter is drawn from a known prior distribution. Even when both the parameter value and the prior distribution are unknown, we may, in practice, utilize partial information available (e.g., occupancy information for some charging stations) to assign prior beliefs for the parameters using this approach (enabling more efficient exploration of the charging stations). In principle, any suitable (positive support) distribution can be used as prior, but for this likelihood and parameter, the gamma distribution is a conjugate prior (meaning that the posterior distribution given observations is also a gamma distribution, and thereby the posterior parameters can be efficiently computed). The prior is then given by

$$P\left(\lambda_u^{\text{queue}}|\alpha_{u,0}^{\text{queue}}, \beta_{u,0}^{\text{queue}}\right) = \text{Gamma}\left(\alpha_{u,0}^{\text{queue}}, \beta_{u,0}^{\text{queue}}\right). \tag{5}$$

Given a sequence of observed queue times $y_1, \ldots, y_t$, the parameters $\alpha_{u,t}^{\text{queue}}$ and $\beta_{u,t}^{\text{queue}}$ of the gamma posterior distribution are given by $\alpha_{u,t}^{\text{queue}} = \alpha_{u,0}^{\text{queue}} + t$ and $\beta_{u,t}^{\text{queue}} = \beta_{u,0}^{\text{queue}} + \sum_{i=1}^{t} y_i$. Similarly, incremental updates of the parameters can be performed using $\alpha_{u,t}^{\text{queue}} = \alpha_{u,t-1}^{\text{queue}} + 1$ and $\beta_{u,t}^{\text{queue}} = \beta_{u,t-1}^{\text{queue}} + y_t$.

### 3.3.2 Charging power model

Ideally, which is also often the case, any given charging station $u \in \mathcal{V}^{\text{feasible}}$ should be able to provide the specified maximum charging power $\varrho_u^{\text{max}}$. Occasionally, however, some charging stations provide less power. Reasons for this may include, e.g., intermittent high load in the surrounding electric grid, limitations of the charging station, etc. Moreover, in this work, we assume that the vehicle is able to fully utilize the charging power provided by the charging station. Again, we note that some of this information may be available to the agent *a priori* and can potentially be used to assign prior beliefs to the parameters.

While a Gaussian model could be sufficient for the anomalous cases described here, it would have to be truncated or rectified to represent the sharp peak in density at $\varrho_u^{\text{max}}$ for a charging station functioning as intended. Then, conjugacy properties may not be used for efficient posterior parameter updates. An alternative, which we describe here, is to use a gamma distribution to model the charging power. In practice, we still rectify the charging power distribution below $\varrho_u^{\text{min}}$ in Section 4 to prevent negative or zero charging power, but this should have a relatively minor impact on the results since the density of the charging power distribution is often concentrated close to $\varrho_u^{\text{max}}$ with the prior distributions that we consider. We define the likelihood function as

$$P\left(\varrho_u^{\text{max}} - \varrho_u^{\text{charge}}|\alpha_u^{\text{charge}}, \beta_u^{\text{charge}}\right) = \text{Gamma}\left(\alpha_u^{\text{charge}}, \beta_u^{\text{charge}}\right). \tag{6}$$

A conjugate prior distribution for both parameters of the gamma likelihood was derived by Damsleth (1975) and further analyzed by Miller (1980), which Damsleth (1975) refers to as the *Gamcon-II prior*. The joint prior distribution over $\alpha_u^{\text{charge}}$ and $\beta_u^{\text{charge}}$ has a set of parameters $\pi_{u,0}^{\text{charge}} > 0$, $\gamma_{u,0}^{\text{charge}} > 0$ and $\xi_{u,0}^{\text{charge}} > 0$, where $\xi_{u,0}^{\text{charge}}\sqrt{\pi_{u,0}^{\text{charge}}} < 1$. Decomposed, the conjugate prior over $\beta_u^{\text{charge}}$ conditional on $\alpha_u^{\text{charge}}$ is also a gamma distribution, defined as

$$P\left(\beta_u^{\text{charge}}|\alpha_u^{\text{charge}}, \gamma_{u,0}^{\text{charge}}, \xi_{u,0}^{\text{charge}}\right) = \text{Gamma}\left(\xi_{u,0}^{\text{charge}} \cdot \alpha_u^{\text{charge}}, \gamma_{u,0}^{\text{charge}}\right). \tag{7}$$

Whereas the prior distribution over $\beta_u^{\text{charge}}$ has a convenient form for both sampling and moment computation, only the unnormalized probability density function for the marginal conjugate prior distribution over $\alpha_u^{\text{charge}}$ is available. It is defined as

$$
\begin{aligned}
P\left(\alpha_u^{\text{charge}} | \pi_{u,0}^{\text{charge}}, \gamma_{u,0}^{\text{charge}}, \xi_{u,0}^{\text{charge}}\right) \propto \exp\Big(\alpha_u^{\text{charge}} \ln \pi_{u,0}^{\text{charge}} &- \xi_{u,0}^{\text{charge}} \alpha_u^{\text{charge}} \ln \gamma_{u,0}^{\text{charge}} \\
&- \xi_{u,0}^{\text{charge}} \ln \Gamma\left(\alpha_u^{\text{charge}}\right) + \ln \Gamma\left(\xi_{u,0}^{\text{charge}} \alpha_u^{\text{charge}}\right)\Big),
\end{aligned}
\tag{8}
$$

where $\Gamma\left(\cdot\right)$ is the well-known gamma function. The joint unnormalized prior distribution over $\alpha_u^{\text{charge}}$ and $\beta_u^{\text{charge}}$ is then the product of Eq. 7 and Eq. 8. With an observed charging power $z_t$, the incremental updates for the parameters of the joint posterior are given by $\pi_{u,t}^{\text{charge}} = \pi_{u,t-1}^{\text{charge}} \cdot (\varrho_u^{\max} - z_t)$, $\gamma_{u,t}^{\text{charge}} = \gamma_{u,t-1}^{\text{charge}} + (\varrho_u^{\max} - z_t)$ and $\xi_{u,t}^{\text{charge}} = \xi_{u,t-1}^{\text{charge}} + 1$. Despite lacking a normalization constant, Eq. 8 can be used to efficiently find the mode of the posterior, since it is log-concave on the (positive real) domain. An unnormalized density function can also be used in adaptive rejection sampling methods to efficiently generate exact samples from the posterior distribution.

## 4  CMAB formulation

We formulate the problem of selecting paths and charging stations through the road network as a sequential decision-making problem under uncertainty. Specifically, we see it as a *combinatorial semi-bandit* (CMAB) problem (Cesa-Bianchi & Lugosi, 2012; Gai et al., 2012), a variant of the classical *multi-armed bandit* (MAB) problem. For a finite horizon $T$, and each iteration $t \in [T]$, the agent has to select and execute an action. In the CMAB setting, this action consists of a subset of objects from a *ground set*. Often, there are constraints on which subsets are allowed to be selected by the agent. The environment gives feedback for each of the objects selected (called *semi-bandit feedback*), the set of which determines the *reward* received by the agent for taking the action.

In our setting, the ground set corresponds to the set of edges in the feasibility graph, i.e., $\mathcal{E}^{\text{feasible}}$. For a source vertex $u^{\text{src}} \in \mathcal{V}^{\text{feasible}}$ and a target vertex $u^{\text{trg}} \in \mathcal{V}^{\text{feasible}}$, fixed for a particular problem instance, the set of allowed actions corresponds to the set of paths $\mathcal{P}_{(u^{\text{src}}, u^{\text{trg}})}^{\text{feasible}}$ from the source to the target in the feasibility graph. In each iteration $t \in [T]$, the agent selects and travels a path $\boldsymbol{p}_t \in \mathcal{P}_{(u^{\text{src}}, u^{\text{trg}})}^{\text{feasible}}$, and receives the path travel time $\tau_{(u,u')}^{\text{path}}$, queue time $\tau_{u'}^{\text{queue}}$ and charging time $\tau_{(u,u')}^{\text{charge}}$ as feedback for each edge $(u,u') \in \boldsymbol{p}_t$. Since the shortest path problem is a minimization problem, we say that an action has a *loss* instead of a reward, where the loss of the traveled path is

$$
L_t(\boldsymbol{p}_t) = \sum_{(u,u') \in \boldsymbol{p}_t} \tau_{(u,u')}^{\text{feasible}},
\tag{9}
$$

where $\tau_{(u,u')}^{\text{feasible}} = \tau_{(u,u')}^{\text{path}} + \tau_{u'}^{\text{queue}} + \tau_{(u,u')}^{\text{charge}}$. Let $\boldsymbol{\theta}$ be an arbitrary vector of model parameters for the entire feasibility graph, where for each vertex $u \in \mathcal{V}^{\text{feasible}}$ we let $\boldsymbol{\theta}_u = \left(\lambda_u^{\text{queue}}, \alpha_u^{\text{charge}}, \beta_u^{\text{charge}}\right)$. Then, we define the *expected loss function* of a path $\boldsymbol{p}$ as

$$
f_{\boldsymbol{\theta}}(\boldsymbol{p}) = \sum_{(u,u') \in \boldsymbol{p}} \left(\tau_{(u,u')}^{\text{path}} + \frac{1}{\lambda_{u'}^{\text{queue}}} + g_{\alpha_{u'}^{\text{charge}}, \beta_{u'}^{\text{charge}}}(u,u')\right),
\tag{10}
$$

where

$$
g_{\alpha_{u'}^{\text{charge}}, \beta_{u'}^{\text{charge}}}(u,u') = \mathbb{E}\left[\varepsilon_{(u,u')}^{\text{path}} / \max\left(\varrho_{u'}^{\min}, \varrho_{u'}^{\text{charge}}\right) \mid \alpha_{u'}^{\text{charge}}, \beta_{u'}^{\text{charge}}\right]
\tag{11}
$$

is the expected charging time, given the parameters $\alpha_{u'}^{\text{charge}}$ and $\beta_{u'}^{\text{charge}}$. Here, the charging power in the denominator is rectified below the minimum charging power $\varrho_{u'}^{\min}$. Throughout this work, $g_{\alpha_{u'}^{\text{charge}}, \beta_{u'}^{\text{charge}}}(u,u')$

---

**Algorithm 1** CMAB charging station selection

---

**Input**: $\alpha_{u,0}^{\text{queue}}, \beta_{u,0}^{\text{queue}}, \pi_{u,0}^{\text{charge}}, \gamma_{u,0}^{\text{charge}}, \xi_{u,0}^{\text{charge}}$

1: **for** $t = 1, \ldots, T$ **do**
2:      **for** $u \in \mathcal{V}^{\text{feasible}}$ **do**
3:          Compute $\hat{\tau}_u^{\text{queue}}, \hat{\alpha}_u^{\text{charge}}$ and $\hat{\beta}_u^{\text{charge}}$ using specified CMAB method and current posterior parameters $\alpha_{u,t-1}^{\text{queue}}, \beta_{u,t-1}^{\text{queue}}, \pi_{u,t-1}^{\text{charge}}, \gamma_{u,t-1}^{\text{charge}}$ and $\xi_{u,t-1}^{\text{charge}}$
4:      **end for**
5:      **for** $(u, u') \in \mathcal{E}^{\text{feasible}}$ **do**
6:          $\hat{\tau}_{(u,u')}^{\text{feasible}} \leftarrow \tau_{(u,u')}^{\text{path}} + \hat{\tau}_{u'}^{\text{queue}} + \hat{g}_{\hat{\alpha}_u^{\text{charge}}, \hat{\beta}_u^{\text{charge}}}(u, u')$
7:      **end for**
8:      $\boldsymbol{p}_t \leftarrow \arg\min_{\boldsymbol{p} \in \mathcal{P}_{(u^{\text{src}}, u^{\text{trg}})}^{\text{feasible}}} \sum_{(u,u') \in \boldsymbol{p}} \hat{\tau}_{(u,u')}^{\text{feasible}}$
9:      **for** each travelled edge $(u, u') \in \boldsymbol{p}_t$ **do**
10:          Observe feedback $\tau_{u'}^{\text{queue}}$ and $\tau_{(u,u')}^{\text{charge}}$
11:          $\alpha_{u',t}^{\text{queue}} \leftarrow \alpha_{u',t-1}^{\text{queue}} + 1$
12:          $\beta_{u',t}^{\text{queue}} \leftarrow \beta_{u',t-1}^{\text{queue}} + \tau_{u'}^{\text{queue}}$
13:          $\varrho_{u'}^{\text{charge}} \leftarrow \frac{\varepsilon_{(u,u')}^{\text{path}}}{\tau_{(u,u')}^{\text{charge}}}$
14:          $\pi_{u',t}^{\text{charge}} \leftarrow \pi_{u',t-1}^{\text{charge}} \cdot \left( \varrho_{u'}^{\max} - \varrho_{u'}^{\text{charge}} \right)$
15:          $\gamma_{u',t}^{\text{charge}} \leftarrow \gamma_{u',t-1}^{\text{charge}} + \left( \varrho_{u'}^{\max} - \varrho_{u'}^{\text{charge}} \right)$
16:          $\xi_{u',t}^{\text{charge}} \leftarrow \xi_{u',t-1}^{\text{charge}} + 1$
17:      **end for**
18: **end for**

---

is approximated as $\hat{g}_{\alpha_{u'}^{\text{charge}}, \beta_{u'}^{\text{charge}}}(u, u')$ using Monte Carlo estimation, by averaging over charging time values computed with samples from the charging power model defined in Eq. 6.

MAB algorithms are usually evaluated using the notion of *regret* until a horizon $T$, which is defined as the sum over all iterations $t \in [T]$ of the difference in expected loss of the best action $\boldsymbol{p}^*$ (defined as in Eq. 1, but for the feasibility graph) and the action $\boldsymbol{p}_t$ selected by the algorithm, such that

$$\text{Regret}(T) = \sum_{t \in [T]} \left( f_{\boldsymbol{\theta}^*}(\boldsymbol{p}_t) - f_{\boldsymbol{\theta}^*}(\boldsymbol{p}^*) \right), \tag{12}$$

where $\boldsymbol{\theta}^*$ is the true underlying parameter vector (in which the parameters of the queue time and charging power distributions are assumed to be drawn from their respective prior distributions). The objective is to find a policy that minimizes the expected regret, where a sub-linear growth with respect to $T$ is generally desired.

## 5 CMAB methods

We adapt three CMAB algorithms for our problem setting: Epsilon-greedy, Thompson Sampling and BayesUCB. For all three algorithms, a shortest path algorithm is used to find the shortest path through the feasibility graph. This is usually called an *oracle* in CMAB literature. While Dijkstra's algorithm is a commonly used oracle for CMABs with shortest path problems (see e.g., Gai et al. 2012; Liu & Zhao 2012; Zou et al. 2014; Åkerblom et al. 2020), we may use the more efficient A* algorithm since the feasibility graph admits a suitable heuristic function. Since the vector of path travel times $\boldsymbol{\tau}^{\text{path}}$ is fixed and known, the direct (beeline) distance between each pair of charging stations can be divided by the maximum allowed speed in the road network (e.g., 120 km/h) to get a value which is guaranteed to underestimate the travel time between those stations. We do not explicitly consider the queue time or charging time in the heuristic function, but clearly, both are non-negative and implicitly underestimated by zero.

All three algorithms follow the same general structure, as outlined in Algorithm 1, which closely corresponds to the CMAB description in Section 4, while also including explicit posterior parameter updates and other details. The primary bottleneck in the computational efficiency of Algorithm 1 is the shortest path computation on the feasibility graph, with a run-time complexity for each iteration $t \in [T]$ of $\mathcal{O}\left( |\mathcal{E}^{\text{feasible}}| + |\mathcal{V}^{\text{feasible}}| \log |\mathcal{V}^{\text{feasible}}| \right)$ if Dijkstra's algorithm is used.

## 5.1 Epsilon-greedy

In each iteration $t \in [T]$, the Epsilon-greedy MAB algorithm selects actions either greedily, according to current parameter estimates of the loss distributions, or uniformly at random. It selects uniform exploration with a small probability $\epsilon_t$ (decreasing with $t$) and greedy otherwise.

In line 3 of Algorithm 1, we retrieve $\hat{\tau}_u^{\text{queue}}$, $\hat{\alpha}_u^{\text{charge}}$ and $\hat{\beta}_u^{\text{charge}}$ through MAP estimation, i.e., by finding the mode of each posterior distribution. This can be done analytically for the gamma prior / posterior in Eq. 5, such that $\hat{\tau}_u^{\text{queue}} \leftarrow \beta_{u,t-1}^{\text{queue}}/(\alpha_{u,t-1}^{\text{queue}} - 1)$, where we assume that $\alpha_{u,t-1}^{\text{queue}} > 1$. For $\hat{\alpha}_u^{\text{charge}}$ and $\hat{\beta}_u^{\text{charge}}$, the Gamcon-II prior / posterior over $\alpha_u^{\text{charge}}$ in Eq. 8 has no analytical formula for the mode, but it can be found numerically. With the mode $\hat{\alpha}_u^{\text{charge}}$, we can calculate $\hat{\beta}_u^{\text{charge}} \leftarrow (\xi_{u,t-1}^{\text{charge}} \cdot \hat{\alpha}_u^{\text{charge}} - 1)/\gamma_{u,t-1}^{\text{charge}}$.

In greedy iterations, the calculated estimates are used directly in line 8 to find a path to travel. In exploration iterations, however, line 8 is changed to provide random exploration of the feasibility graph. Chen et al. (2013) (supplementary material) introduced a CMAB version of Epsilon-greedy, which we adapt here. First, a vertex $u^{\text{rand}} \in \mathcal{V}^{\text{feasible}}$ is selected uniformly at random. Then, we find the paths $\boldsymbol{p}_t^{(1)} \leftarrow \arg\min_{\boldsymbol{p} \in \mathcal{P}_{(u^{\text{src}}, u^{\text{rand}})}^{\text{feasible}}} \sum_{(u,u') \in \boldsymbol{p}} \hat{\tau}_{(u,u')}^{\text{feasible}}$ and $\boldsymbol{p}_t^{(2)} \leftarrow \arg\min_{\boldsymbol{p} \in \mathcal{P}_{(u^{\text{rand}}, u^{\text{trg}})}^{\text{feasible}}} \sum_{(u,u') \in \boldsymbol{p}} \hat{\tau}_{(u,u')}^{\text{feasible}}$, which we concatenate to get $\boldsymbol{p}_t$.

## 5.2 Thompson Sampling

Thompson Sampling (Thompson, 1933) is one of the oldest MAB algorithms, which has recently been adapted to CMAB problems (Wang & Chen, 2018), including shortest path problems with stochastic edge weights for various applications (Wang & Chen, 2018; Åkerblom et al., 2020; 2023b). Like Epsilon-greedy, it performs randomized exploration, but it does so in every iteration and in a more guided way. It utilizes the knowledge encoded in the prior and posterior distributions, by sampling paths according to the probability that they are optimal (given the prior beliefs and the observations from the environment).

In Algorithm 1, only line 3 needs to be adapted to this method. Here, the expected queue time $\hat{\tau}_u^{\text{queue}}$ and charging power parameters $\hat{\alpha}_u^{\text{charge}}$ and $\hat{\beta}_u^{\text{charge}}$ are calculated using parameters sampled from the current posterior distributions. For the queue time prior distribution in Eq. 5, sampling the rate parameter $\hat{\lambda}_u^{\text{queue}}$ from the gamma distribution is straightforward, which gives an expected queue time of $\hat{\tau}_u^{\text{queue}} \leftarrow 1/\hat{\lambda}_u^{\text{queue}}$. Similar to the mode calculations in Section 5.1, sampling $\hat{\alpha}_u^{\text{charge}}$ from the Gamcon-II prior and posterior distributions is not as convenient. However, we can utilize *adaptive rejection sampling* (ARS) (Gilks & Wild, 1992) to generate exact posterior samples, since it only requires a (log-concave, but not necessarily normalized) probability density function, like Eq. 8. In this work, we specifically use an extension called *transformed density rejection* (TDR) (Hörmann, 1995). Once a sample of $\hat{\alpha}_u^{\text{charge}}$ is obtained, the conditional gamma prior distribution in Eq. 7 can be used to sample $\hat{\beta}_u^{\text{charge}}$.

## 5.3 BayesUCB

MAB algorithms based on *upper confidence bounds* (UCB) (Auer, 2002) use high probability overestimates of actions' expected rewards to explore the environment. By doing this, UCB methods follow the principle of *optimism in the face of uncertainty* to select promising actions. UCB methods have been shown to have good performance in many different problem settings, and have been adapted to CMAB settings (Chen et al., 2013), including shortest path problems.

We adapt a Bayesian version of UCB called *BayesUCB* (Kaufmann et al., 2012a; Kaufmann, 2018) to this setting so that we can utilize the prior distributions for exploration. Again, like for Thompson Sampling, we only have to modify line 3 of Algorithm 1 to implement this method. BayesUCB uses (lower, in this case) quantiles of the posterior distributions over expected action losses as optimistic estimates. Given a probability distribution $\chi$ and a probability $\nu$, the quantile function $Q(\nu, \chi)$ is defined such that $\Pr_{x \sim \chi} \{x \leq Q(\nu, \chi)\} = \nu$. For the queue time $\hat{\tau}_u^{\text{queue}}$, a high rate parameter $\hat{\lambda}_u^{\text{queue}}$ results in a low expected travel time $\hat{\tau}_u^{\text{queue}} \leftarrow 1/\hat{\lambda}_u^{\text{queue}}$. Hence, an upper quantile of Eq. 5 should be sought, i.e.,

$$\hat{\lambda}_u^{\text{queue}} \leftarrow Q\left(1 - 1/t, P\left(\lambda_u^{\text{queue}} | \alpha_{u,t-1}^{\text{queue}}, \beta_{u,t-1}^{\text{queue}}\right)\right) , \tag{13}$$

where we use the probability value $(1 - 1/t)$ suggested by Kaufmann et al. (2012a). Since the Gamcon-II prior and posterior distributions do not admit a convenient way of computing quantile values, we settle on using the mode of Eq. 8 to obtain $\hat{\alpha}_u^{\text{charge}}$. However, we utilize the mode $\hat{\alpha}_u^{\text{charge}}$ to compute an upper confidence bound for $\beta_u^{\text{charge}}$ using Eq. 7, such that

$$\hat{\beta}_u^{\text{charge}} \leftarrow Q\left(1 - 1/t, P\left(\beta_u^{\text{charge}} | \hat{\alpha}_u^{\text{charge}}, \gamma_{u,t-1}^{\text{charge}}, \xi_{u,t-1}^{\text{charge}}\right)\right) . \tag{14}$$

Then, as before, we can calculate an optimistic (low) estimate of the mean charging time.

## 6 Experiments

To evaluate the feasibility graph construction procedure described in Section 3 and the CMAB methods outlined in Section 5, we perform realistic experiments in country-sized road networks. We define three different problem instances, characterized by their origins and destinations, across the northern European countries of Sweden, Norway and Finland. We utilize open datasets for the road network map data (OpenStreetMap contributors, 2022) and the charging station data (Open Charge Map, 2022) of each country.

### 6.1 Energy consumption and travel time

For the vehicle energy consumption, we use a simplified vehicle longitudinal dynamics model based on Guzzella et al. (2007), where we only consider the maximum speed of each road segment in the road network, i.e., disregard accelerations, decelerations and altitude changes. The vehicle parameters that we use are (arbitrarily) for a medium duty truck, similar to the one used by Åkerblom et al. (2020). For an edge $e \in \mathcal{E}^{\text{road}}$, the model is defined as

$$\varepsilon_e^{\text{road}} = \frac{mgC_r d_e^{\text{road}} + 0.5 C_d A \rho d_e^{\text{road}} (v_e^{\text{road}})^2}{3600\eta}, \tag{15}$$

where $m$ is the vehicle mass (13700 kg), $g$ is the gravitational acceleration (9.81 m/s$^2$), $C_r$ is the rolling resistance coefficient (0.0064, assumed to be the same for the entire road network), $d_e^{\text{road}}$ is the length of the road segment (m), $C_d$ is the air drag coefficient of the vehicle (0.7), $A$ is the frontal surface area of the truck (8 m$^2$), $\rho$ is the air density (1.2 kg/m$^3$, assumed to be the same everywhere), $v_e^{\text{road}}$ is the maximum speed of the road segment (m/s), and $\eta$ is the battery-to-wheel energy conversion efficiency (assumed to perfect, i.e., 1). The battery capacity ($2.5 \cdot 10^8$ Ws $\approx 69.4$ kWh) of the vehicle is assigned to be very low, so that it is required to charge often. The travel time of the edge is assumed to be $\tau_e^{\text{road}} = d_e^{\text{road}}/v_e^{\text{road}}$.

### 6.2 Experimental setup

First, each of the country road network graphs is transformed into a feasibility graph according to the procedure described in Section 3. For simplicity, we remove all charging stations with lower specified power than 10 kW, since slower charging stations should be less relevant for long-distance travel. Furthermore, we assume that each charging location has a single charging station (by removing all except the one with the highest specified charging power, as well as any duplicates). Table 1 shows the number of vertices and edges in the initial road graph $\mathcal{G}^{\text{road}}\left(\mathcal{V}^{\text{road}}, \mathcal{E}^{\text{road}}\right)$ and constructed feasibility graph $\mathcal{G}^{\text{feasible}}\left(\mathcal{V}^{\text{feasible}}, \mathcal{E}^{\text{feasible}}\right)$ of each problem instance. Figures 1a, 1c and 1e visualize all edges of the road network graphs of Sweden, Norway and Finland, respectively, as well as examples of the explored paths and charging stations visited when Thompson Sampling is applied to the problem. Figures 1b, 1d and 1f show the corresponding feasibility graphs for each of the networks. In the feasibility graphs, we can see that some parts of the road networks are unreachable from the rest of the road network, given the specified battery capacity of 69.4 kWh.

For each vertex $u \in \mathcal{V}^{\text{feasible}}$ and the queue time prior distribution defined in Eq. 5, we assign the prior parameters as $\alpha_{u,0}^{\text{queue}} = 2$ and $\beta_{u,0}^{\text{queue}} = 2400$, and for the charging power prior distribution in Eq. 7 and Eq.

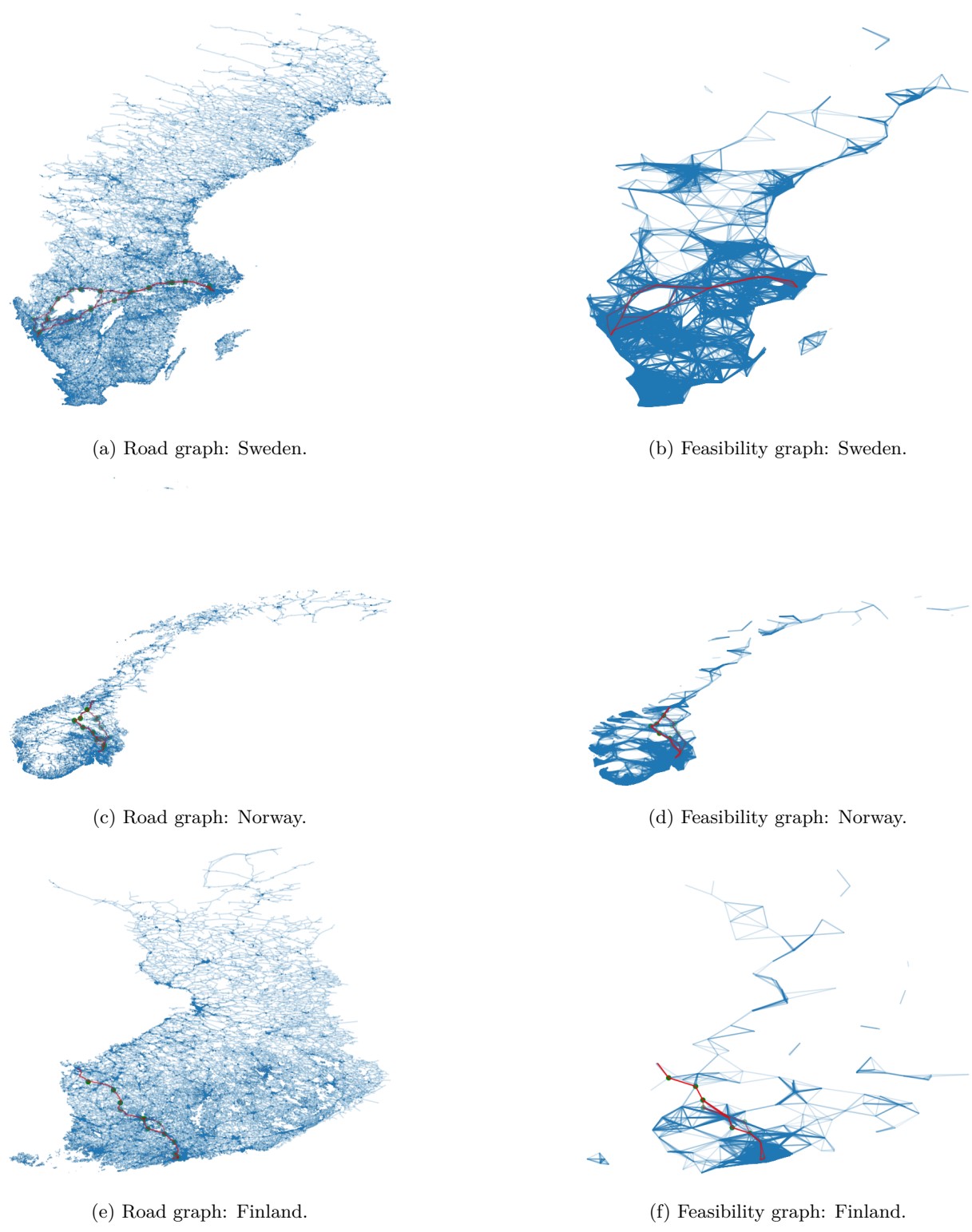

Figure 1: Road and feasibility graphs for each of the problem instances in blue, with Thompson Sampling exploration of paths in red and charging stations in green, where the opacity indicates degree of exploration for both.

| Instance | $|\mathcal{V}^{\text{road}}|$ | $|\mathcal{E}^{\text{road}}|$ | $|\mathcal{V}^{\text{feasible}}|$ | $|\mathcal{E}^{\text{feasible}}|$ |
|---|---|---|---|---|
| Sweden | $6.8 \cdot 10^5$ | $1.5 \cdot 10^6$ | $1.7 \cdot 10^3$ | $1.6 \cdot 10^5$ |
| Norway | $3.6 \cdot 10^5$ | $7.6 \cdot 10^5$ | $1.1 \cdot 10^3$ | $8.5 \cdot 10^4$ |
| Finland | $4.3 \cdot 10^5$ | $9.5 \cdot 10^5$ | $5.5 \cdot 10^2$ | $7.0 \cdot 10^4$ |

Table 1: Sizes of the road and feasibility graphs for all problem instances.

8, we set the parameters so that $\pi_{u,0}^{\text{charge}} = \exp(13.5)$, $\gamma_{u,0}^{\text{charge}} = 300$ and $\xi_{u,0}^{\text{charge}} = 3$. Furthermore, we scale the samples and expected values of the gamma distribution in Eq. 6 by 300 to achieve a sufficiently high charging power variance. For the rectification of the charging power below $\varrho_u^{\text{min}}$, we assume $\varrho_u^{\text{min}} = \varrho_u^{\text{max}}/2$. We choose $N = 1000$, and can then estimate the expected charging time using Monte Carlo sampling, such that

$$\hat{g}_{\alpha_{u'}^{\text{charge}},\beta_{u'}^{\text{charge}}}(u, u') = \frac{1}{N} \sum_{k=1}^{N} \frac{\varepsilon_{(u,u')}^{\text{path}}}{\max\left(\varrho_u^{\text{min}}, \varrho_u^{\text{max}} - 300 z_k\right)} , \tag{16}$$

where $z_k \sim \text{Gamma}\left(\alpha_{u'}^{\text{charge}}, \beta_{u'}^{\text{charge}}\right)$.

For Epsilon-Greedy, we let $\epsilon_t = 1/\sqrt{t}$. Moreover, for Thompson Sampling, we experience that TDR occasionally fails to produce samples when the posterior distribution over $\alpha_u^{\text{charge}}$ gets too concentrated, typically after a few hundred observations. When this happens, we switch to the mode of the distribution for the specific charging station $u$, while continuing posterior sampling for all other charging stations. Besides Epsilon-Greedy (E-GR), Thompson Sampling (TS) and BayesUCB (B-UCB), we also include a pure greedy method (GR) in the experiments (i.e., Epsilon-Greedy with $\epsilon_t = 0$), as a baseline.

### 6.3 Results

We run all experiments with a horizon $T = 1000$. We include the following problem instances: Sweden (Gothenburg to Stockholm), Norway (Oslo to Trondheim) and Finland (Helsinki to Vaasa). For all pairs of problem instances and CMAB methods, we run the same experiment 10 times, with different random seeds. The regret results are summarized in Table 2, as well as through regret plots with standard error regions in Figures 2a, 2b and 2c.

| Method | Sweden | Norway | Finland |
|---|---|---|---|
| GR | $1.8 \cdot 10^6 (\pm 1.3 \cdot 10^6)$ | $6.2 \cdot 10^5 (\pm 8.6 \cdot 10^5)$ | $1.8 \cdot 10^6 (\pm 1.7 \cdot 10^6)$ |
| E-GR | $4.4 \cdot 10^6 (\pm 5.4 \cdot 10^5)$ | $3.2 \cdot 10^6 (\pm 1.1 \cdot 10^6)$ | $2.3 \cdot 10^6 (\pm 9.2 \cdot 10^5)$ |
| TS | $4.0 \cdot 10^5 (\pm 2.7 \cdot 10^5)$ | $2.2 \cdot 10^5 (\pm 2.2 \cdot 10^5)$ | $4.4 \cdot 10^5 (\pm 1.9 \cdot 10^5)$ |
| B-UCB | $7.6 \cdot 10^5 (\pm 2.4 \cdot 10^5)$ | $2.7 \cdot 10^5 (\pm 1.3 \cdot 10^5)$ | $5.2 \cdot 10^5 (\pm 3.3 \cdot 10^5)$ |

Table 2: Final average ($\pm$ standard deviation) of regret at iteration $T = 1000$ for all problem instances.

In Table 3, we report the average and standard deviation of the per-iteration run-time (in seconds) of each method and problem instance. The CMAB methods are implemented with Python using the SciPy library, for implementations of statistical functions, including the Transformed Density Rejection method (Hörmann, 1995), and the NetworkX library (Hagberg et al., 2008), for the implementation of the A* algorithm. All run-time measurements were performed on a single core of a laptop with an Intel(R) Core(TM) i7-10850H CPU (2.70 GHz) and 32.00 GB RAM.

In general, the Epsilon-Greedy method incurs the highest final regret of all the methods, for all problem instances. This can be explained by the uniformly random selection of charging stations, which means that the method takes very long detours, sometimes to the other side of the country. Following close behind is

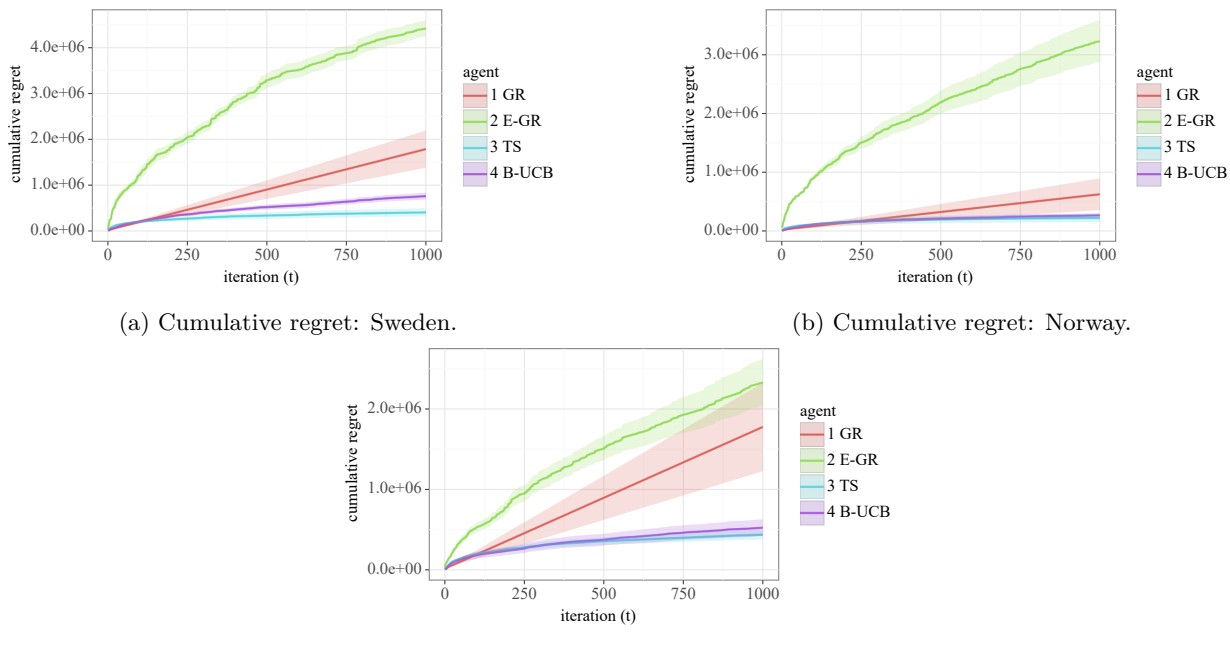

(a) Cumulative regret: Sweden.

(b) Cumulative regret: Norway.

(c) Cumulative regret: Finland.

Figure 2: Plots of cumulative regret averaged over 10 runs (with standard error regions), as a function of the iteration $t$, for each of the problem instances, and the CMAB methods Greedy (GR), Epsilon-Greedy (E-GR), Thompson Sampling (TS) and BayesUCB (B-UCB).

| Method | Sweden | Norway | Finland |
|--------|--------|--------|---------|
| GR | 2.04($\pm$0.18) | 2.61($\pm$0.27) | 3.09($\pm$0.37) |
| E-GR | 2.80($\pm$0.51) | 2.83($\pm$0.79) | 3.38($\pm$0.77) |
| TS | 2.79($\pm$0.78) | 2.92($\pm$0.62) | 2.93($\pm$0.36) |
| B-UCB | 3.47($\pm$0.32) | 2.97($\pm$0.37) | 3.46($\pm$0.42) |

Table 3: Average (over 1000 iterations) per-iteration run-time (s) ($\pm$ standard deviation) of each CMAB method, for different problem instances.

the Greedy method, which quickly converges to sub-optimal paths. This becomes even more apparent in the regret plots, which show the regret (averaged over 10 runs) as a function of the iteration $t$. For each of the problem instances, the Greedy method exhibits a linear increase in regret, while the other methods (even Epsilon-greedy) continuously find better paths. For all problem instances, Thompson Sampling performs slightly better than BayesUCB, which may be due to the more optimistic exploration of BayesUCB resulting in a wider spread of explored paths (and consequently, occasionally more regret incurred).

# 7 Discussion and conclusion

The key benefit of the feasibility graph structure is to reduce the number of unknown parameters that the learning agent needs to estimate. While the set of base arms in CMAB online shortest path problems often correspond to the set of edges (see e.g., Åkerblom et al. 2023a), in this case, the total number of charging stations is orders of magnitude lower than the paths between them (see Table 1). Since the worst-case expected regret of CMAB methods usually scales with the number of base arms (Chen et al., 2013; Wang & Chen, 2018), keeping this number low is desirable. Even with the extension for partial charging mentioned in Section 3.2, the number of base arms would stay the same (ensuring scalability).

While the assumptions in this work are relatively strong (e.g., the assumption of mutually independent charging time and queue time with fixed distributions, or the M/M/1 queue assumption), we emphasize that, in principle, the feasibility graph structure enables more expressive models associated with each charging station. One such possible extension of this work can be to take accessible real-time information on the congestion levels of surrounding roads into account (which both the charging time and queue time likely depend on), by casting the problem as a CMAB version of the contextual MAB problem (Lu et al., 2010).

In conclusion, this work introduces a novel and extensible combinatorial semi-bandit framework for navigation and charging station selection in road networks where the queue time and charging power of each charging station are stochastic with unknown distributions, the parameters of which are generated from known prior distributions. We utilize conjugate prior distributions for the exponential and gamma models to estimate the loss distributions and induce exploration. Finally, we demonstrate the performance of our framework on several country-sized road and charging networks.

### Acknowledgments

This work is funded by the Strategic Vehicle Research and Innovation Programme (FFI) of Sweden, through the project EENE (reference number: 2018-01937). Map data copyrighted OpenStreetMap contributors and available from `https://www.openstreetmap.org`. The structure of our simulation framework was originally based on the code of Russo et al. (2018), though has since been completely rewritten.

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

# A   Appendix

| Notation | Description |
| --- | --- |
| $\mathcal{G}^{\text{road}}$ | Road graph |
| $\mathcal{V}^{\text{road}}$ | Road graph vertices |
| $\mathcal{E}^{\text{road}}$ | Road graph edges |
| $\boldsymbol{\tau}^{\text{road}}$ | Vector of road graph edge travel times |
| $\tau_e^{\text{road}}$ | Road graph travel time of edge $e$ |
| $\tau_{\boldsymbol{p}}^{\text{road}}$ | Road graph travel time of path $\boldsymbol{p}$ |
| $\varepsilon_e^{\text{road}}$ | Road graph energy consumption of edge $e$ |
| $\varepsilon_{\boldsymbol{p}}^{\text{road}}$ | Road graph energy consumption of path $\boldsymbol{p}$ |
| $\mathcal{V}^{\text{charge}}$ | Charging station vertices |
| $\varrho_u^{\text{charge}}$ | Charging station power (actual) of vertex $u$ |
| $\varrho_u^{\text{max}}$ | Charging station maximum power of vertex $u$ |
| $\varrho_u^{\text{min}}$ | Charging station minimum power of vertex $u$ |
| $u$ | A vertex |
| $u'$ | A vertex (alternative) |
| $u^{\text{src}}$ | Source vertex |
| $u^{\text{trg}}$ | Target vertex |
| $e$ | An edge |
| $\boldsymbol{p}$ | A path |
| $\boldsymbol{p}^*$ | Shortest path (implicitly between specified source and target vertices) |
| $\boldsymbol{p}^*_{(u,u')}$ | Shortest path between vertices $u$ and $u'$ |
| $\mathcal{P}^{\text{road}}_{(u^{\text{src}},u^{\text{trg}})}$ | Set of paths in road graph between source and target vertices |
| $\mathcal{G}^{\text{feasible}}$ | Feasibility graph |
| $\mathcal{V}^{\text{feasible}}$ | Feasibility graph vertices |
| $\mathcal{E}^{\text{feasible}}$ | Feasibility graph edges |
| $\mathcal{E}^{\text{path}}$ | Edges corresponding to shortest paths between charging stations in road graph |
| $\tau^{\text{path}}_{(u,u')}$ | Path travel time between vertices $u$ and $u'$ |
| $\tau^{\text{charge}}_{(u,u')}$ | Charging time required for travel between vertices $u$ and $u'$ |
| $\tau_u^{\text{queue}}$ | Queue time required for charging station at vertex $u$ |
| $\tau_e^{\text{feasible}}$ | Feasibility graph travel time of edge $e$ |
| $\varepsilon^{\text{path}}_{(u,u')}$ | Path energy consumption between vertices $u$ and $u'$ |
| $\varepsilon^{\text{max}}$ | Maximum battery capacity of vehicle |
| $\varepsilon^{\text{min}}$ | Minimum battery capacity of vehicle |
| $\lambda_u^{\text{queue}}$ | Queue time distribution rate parameter of vertex $u$ |
| $\alpha_{u,0}^{\text{queue}}$ | Shape prior parameter for queue time rate parameter, of vertex $u$ |
| $\beta_{u,0}^{\text{queue}}$ | Rate prior parameter for queue time rate parameter, of vertex $u$ |
| $\alpha_{u,t}^{\text{queue}}$ | Shape posterior parameter for queue time rate parameter, at time $t$ and vertex $u$ |
| $\beta_{u,t}^{\text{queue}}$ | Rate posterior parameter for queue time rate parameter, at time $t$ and vertex $u$ |
| $\alpha_u^{\text{charge}}$ | Charging power distribution shape parameter of vertex $u$ |
| $\beta_u^{\text{charge}}$ | Charging power distribution rate parameter of vertex $u$ |
| $\pi_{u,0}^{\text{charge}}$ | First prior parameter for charging power parameters, of vertex $u$ |
| $\gamma_{u,0}^{\text{charge}}$ | Second prior parameter for charging power parameters, of vertex $u$ |
| $\xi_{u,0}^{\text{charge}}$ | Third prior parameter for charging power parameters, of vertex $u$ |
| $\pi_{u,t}^{\text{charge}}$ | First posterior parameter for charging power parameters, at time $t$ and vertex $u$ |
| $\gamma_{u,t}^{\text{charge}}$ | Second posterior parameter for charging power parameters, at time $t$ and vertex $u$ |
| $\xi_{u,t}^{\text{charge}}$ | Third posterior parameter for charging power parameters, at time $t$ and vertex $u$ |

Table 4: Summary of the notation used throughout the paper.

| Notation | Description |
|---|---|
| $t$ | Time step |
| $T$ | Time horizon |
| $\boldsymbol{p}_t$ | Path (action) selected by CMAB algorithm at time $t$ |
| $L_t(\boldsymbol{p})$ | Loss of path $\boldsymbol{p}$ at time $t$ |
| $\boldsymbol{\theta}$ | Vector of all parameters of feasibility graph |
| $\boldsymbol{\theta}^*$ | True underlying (unknown) vector of all parameters of feasibility graph |
| $\boldsymbol{\theta}_u$ | Vector of parameters $\left(\lambda_u^{\text{queue}}, \alpha_u^{\text{charge}}, \beta_u^{\text{charge}}\right)$ of vertex $u$ |
| $f_{\boldsymbol{\theta}}(\boldsymbol{p})$ | Expected loss function, w.r.t. parameter vector $\boldsymbol{\theta}$, applied to path $\boldsymbol{p}$ |
| $g_{\alpha,\beta}(u, u')$ | Expected charging time of edge $(u, u')$, given parameters $\alpha$ and $\beta$ |
| $\hat{g}_{\alpha,\beta}(u, u')$ | Estimated charging time of edge $(u, u')$, given parameters $\alpha$ and $\beta$ |
| $\text{Regret}(T)$ | Regret of CMAB algorithm at horizon $T$ |
| $\epsilon_t$ | Exploration probability of the Epsilon-greedy CMAB algorithm |
| $\hat{\tau}_u^{\text{queue}}$ | Estimate of mean queue time, for vertex $u$ |
| $\hat{\lambda}_u^{\text{queue}}$ | Estimate of queue time rate parameter, for vertex $u$ |
| $\hat{\alpha}_u^{\text{charge}}$ | Estimate of charging power shape parameter, for vertex $u$ |
| $\hat{\beta}_u^{\text{charge}}$ | Estimate of charging power rate parameter, for vertex $u$ |
| $Q\left(\nu, \chi\right)$ | Quantile function for distribution $\chi$ and probability value $\nu$ |
| $m$ | Vehicle mass |
| $g$ | Gravitational acceleration |
| $C_r$ | Rolling resistance coefficient |
| $d_e^{\text{road}}$ | Length of the road segment, for edge $e$ |
| $C_d$ | Air drag coefficient |
| $A$ | Frontal surface area of vehicle |
| $v_e^{\text{road}}$ | Maximum speed of the road segment, for edge $e$ |
| $\rho$ | Air density |
| $\eta$ | Battery-to-wheel energy conversion efficiency |

Table 4: Summary of the notation used throughout the paper.

