# OpenReview forum: "A Combinatorial Semi-Bandit Approach to Charging Station Selection for Electric Vehicles"
_TMLR — Accepted by TMLR_

### Review · Reviewer_TZzB · 2023-09-11

**Summary Of Contributions:**

This paper presents an adaptation of a classical online routing problem to incorporate charging delays. As for the classical routing problem, the problem is viewed as a combinatorial bandit problem where agent choose the set of routes. The paper contains mostly three contributions:
1. The modelization of the problem
2. How to adapt classical learning algorithm (UCB variants and Thompson sampling) to this problem
3. An experimental validation.

**Audience:**

Yes

**Broader Impact Concerns:**

This work is mostly a theoretical work and I do not foresee any broader impact problems.

**Claims And Evidence:**

Yes

**Requested Changes:**

The main change requested concern my weakness 1. The model need not necessarily be changed by the discussion should be rewritten.

**Strengths And Weaknesses:**

Strengths:
- routing problem with energy constraints is becoming a relevant problem with the deployment of electric transportation.
- the paper is well written and clearly presented.
- Several variants of learning algorithms are presented.

Weaknesses:
1. What is meant by "queuing delay" is not clear. In particular, the justification of this queueing delay by an M/M/1 model seems very far-fetched.
2. (linked to the previous point): congestion is not modeled.
3. The problem is a minor variation of a combinatorial bandit problem where the energy constraint is added.
4. Is it not clear if the studied problem corresponds to a realistic model for two reasons:
   - in practice, one would probably have prior knowledge of delays and/or charging times, which make the learning problem less relevant
   - the case when the delays or charging time are unknown would probably be when there is congestion, in which case the problem would have time-varying parameters (that are not considered here).
5. The novelty is somehow limited as it is a variation of the classical online routing problem.

---

> ### Author Response · Authors · 2023-09-26
> **Response to Reviewer TZzB**
>
> Thank you for your review and helpful comments / suggestions! We have attempted to address the comments in the revised manuscript and below:
>
> **1:** We now clarify what we mean by queue time in the beginning of Section 3, i.e. the time spent waiting for a currently occupied charging station, and also describe how observations of the queue time might be obtained, in the first paragraph of Section 3.3. While we agree that the assumptions in this work are relatively strong (e.g., the assumption of mutually independent charging time and queue time with fixed distributions, or the M/M/1 queue assumption), we emphasize that, in principle, the feasibility graph structure enables more expressive models associated with each charging station. We have added this comment to Section 7, which is rewritten.
>
> **2:** We now write in Section 7 that one possible extension of this work can be to take accessible real-time information on the congestion levels of surrounding roads into account (which both the charging time and queue time likely depend on), by casting the problem as a combinatorial version of the contextual MAB problem.
>
> **3:** We consider a main contribution of this work to be the way that the feasibility graph is constructed to allow application of bandit methods to this large-scale problem. In particular, which we now emphasise in Section 7, the key benefit of the feasibility graph structure is to reduce the number of unknown parameters that the learning agent needs to estimate. While the set of base arms in CMAB online shortest path problems often correspond to the set of edges (see e.g., Åkerblom et al. 2023a), in this case, the total number of charging stations is orders of magnitude lower than the paths between them (see Table 1). Since the worst-case expected regret of CMAB methods usually scales with the number of base arms (Chen et al., 2013; Wang & Chen, 2018), keeping this number low is desirable. Even with the extension for partial charging mentioned in Section 3.2, the number of base arms would stay the same (ensuring scalability).
>
> **4.1:** In Sections 3.3.1 and 3.3.2 we now further motivate the problem (and the Bayesian approach). This is method is specifically for when the we have limited access to information, which might be due to a large number of charging station operators, differences in the type of information they provide, etc. By using this approach, even when both the parameter value and the prior distribution are unknown, we may, in practice, utilize partial information available (e.g., occupancy information for some charging stations) to assign prior beliefs for the parameters (enabling more efficient exploration of the charging stations).
>
> **4.2:** See the answer to "Weakness 2" above.
>
> **5:** See the answer to "Weakness 3" above.

---

### Review · Reviewer_243U · 2023-09-12

**Summary Of Contributions:**

This work addresses the problem of long-distance navigation for battery electric vehicles, where one or more charging sessions are required to reach the intended destination. The authors propose a combinatorial semi-bandit framework for exploring the road network to learn the parameters of the queue time and charging power distributions. They first present a method for transforming the road network graph into a graph of feasible paths. Then, they develop a Bayesian approach to model the stochastic edge weights, utilizing conjugate priors for the one-parameter exponential and two-parameter gamma distributions. Finally, they use combinatorial versions of Thompson Sampling, BayesUCB and epsilon-greedy to solve the problem. They provide experimental results on real-world road network datasets.

**Audience:**

Yes

**Broader Impact Concerns:**

I do not have ethical concerns.

**Claims And Evidence:**

Yes

**Requested Changes:**

Please see the review on weaknesses above.

**Strengths And Weaknesses:**

**Strengths:**

1.	The considered problem, long-distance navigation for battery electric vehicles, is very interesting and important in transportation applications. The proposed formulation, especially the formulation of travel time, queue time and charging time, is novel and effectively characterizes the time cost of battery electric vehicles.
2.	This work provides extensive experimental results on real-world country-sized data, including road network data for Norway, Sweden and Finland. This demonstrates the practical performance of the proposed algorithmic framework, and makes a nice empirical contribution to the multi-armed bandit community.

**Weaknesses:**

1.	It seems that this work does not provide theoretical analysis for the proposed algorithms, e.g., the regret analysis, which is important for multi-armed bandit problems.
2.	The combinatorial bandit part in the proposed algorithmic framework seems a straightforward application of existing algorithms, i.e., epsilon-greedy, Thompson Sampling and BayesUCB, which reduces the novelty of algorithm design.
3.	Is the experiment open-source?

---

> ### Author Response · Authors · 2023-09-26
> **Response to Reviewer 243U**
>
> Thank you for the review and the helpful comments! We attempt to address them in the revised version:
>
> **"It seems that this work does not provide theoretical analysis for the proposed algorithms, e.g., the regret analysis, which is important for multi-armed bandit problems."**
>
> Though we do not currently include an upper bound on the expected regret for any of the methods applied to this problem, we now more clearly specify how we adapt the construction of the feasibility graph to the way in which the expect regret of combinatorial bandit methods typically scale with the number of base arms. Regarding a regret bound, however, for Thompson Sampling specifically, it should be possible to combine a technique used to analyse UCB for MAB with sub-exponential distributions (Jia et al., 2021) with the Thompson Sampling analysis by (Russo & van Roy, 2014) to derive a Bayesian Regret bound, and extend it to the combinatorial setting like (Åkerblom et al., 2023a). We expect an upper bound derived in this way to be consistent with existing results.
>
> **"The combinatorial bandit part in the proposed algorithmic framework seems a straightforward application of existing algorithms, i.e., epsilon-greedy, Thompson Sampling and BayesUCB, which reduces the novelty of algorithm design."**
>
> We consider a main contribution of this work to be the way that the feasibility graph is constructed to allow application of bandit methods to this large-scale problem. In particular, which we now emphasise in Section 7, the key benefit of the feasibility graph structure is to reduce the number of unknown parameters that the learning agent needs to estimate. While the set of base arms in CMAB online shortest path problems often correspond to the set of edges (see e.g., Åkerblom et al. 2023a), in this case, the total number of charging stations is orders of magnitude lower than the paths between them (see Table 1). Since the worst-case expected regret of CMAB methods usually scales with the number of base arms (Chen et al., 2013; Wang & Chen, 2018), keeping this number low is desirable. Even with the extension for partial charging mentioned in Section 3.2, the number of base arms would stay the same (ensuring scalability).
>
> **"Is the experiment open-source?"**
>
> It is not currently open-source, but we plan to release it as such after the paper is published.

---

### Review · Reviewer_r8md · 2023-09-16

**Summary Of Contributions:**

In this study, they develop an online self-learning algorithmic framework for BEV navigation. They view the task as a sequential decision-making problem under uncertainty, between exploring charging stations to learn more information about them and using previously collected knowledge to select charging stations that are likely to be good charging stations. To address the trade-off, they model it as a combinatorial semi-bandit problem. Within this framework, this paper adopts a Bayesian approach with conjugation that is new in the bandit literature.

**Audience:**

Yes

**Broader Impact Concerns:**

I do not see a negative social impact by this work.

**Claims And Evidence:**

No

**Requested Changes:**

Please see the major and minor issues above.

**Strengths And Weaknesses:**

Strengths:
1. This study is the first to address the challenging real-world problem of charging station selection in a partial information setting using a multi-armed bandit approach.

2. This work gives an example of a large-scale real-world application of multi-armed bandit. To achieve such scalability, they introduce the transformation of the road graph into a feasibility graph and precompute feasible paths between charging stations to improve run-time efficiency.

Concerning point 2, I believe this work is a very interesting and important adaption of combinatorial bandits to solve real-world problems.

However, there are some issues to be considered in the revised version.

Major issue:
１：Presentaion

There needs to be a clear separation between the definition of notations and quantities and the formulation of the problem, such as what is given as the bandit problem.
In other words, in Section 4: CMAB formulation, please be more explicit about which parameters are known, and for which parameters, when and how the observations are obtained, and the assumptions about the observations.

Presentation for the static problem formulation also has issues:
Section 3.1 merely defines the path problem in Equation 1. Charging stations and energy consumption are not relevant here. Or did you use quantities related to charging stations and associated energy consumption in the modeling of Equation 1?

In the paragraph "The general resource-constrained shortest path problem (Joksch, 1966) (with energy as the resource) is still computationally hard, especially when resource ...."
Could you please specify the exact assumptions when you refer "with these assumptions", to consider computational efficiency? Without a definition of  “resource-constrained shortest path problem”, the statement of this graph is unclear.

Because of the above
It is very unclear to understand the "3.3 Probabilistic queue and charging times" formulation.


2: Why a Bayesian approach is more desired here?


It is unclear why a Bayesian approach was taken to solve this problem, whereas the frequency-based approach is a standard solution in CMAB literature. In particular, [Chen 2013] proposed the CUCB algorithm that can be used with approximate oracles, and subsequent studies propose a variant of CUCB for extensions. HOWEVER, unless I am missing something, this study does not apply CUCB to solve their problem, and it is unclear why.


Minor issue:

1.What is the domein for \varepsilon_e^{road} and \tau_2^{road}? Is it in \mathbb{R} or \mathbb{N}?

2. Which set of location of charging stations \mathcal{V}^{charge} are known to the agent?

3. \varrho^{max} and \varrho^{min} are known to the agent?

4. Are a maximum battery capacity ε_max and minimum battery capacity ε_min are known to the agent?

5. Can you write this assumption “we assume that each charging session has to fully charge the battery.” with mathematical formulation?


6. Weighted directed graphs can be given in the form G=(V,E,w). Superscripts such as {road} reduce readability.

---

> ### Author Response · Authors · 2023-09-26
> **Response to Reviewer r8md**
>
> Thank you for your detailed review! In the revised version, we attempt to address the issues that you have raised. First, regarding the presentation:
>
> **"There needs to be a clear separation between the definition of notations and quantities and the formulation of the problem, such as what is given as the bandit problem. In other words, in Section 4: CMAB formulation, please be more explicit about which parameters are known, and for which parameters, when and how the observations are obtained, and the assumptions about the observations."**
>
> We have attempted to clarify this, but earlier, in the beginning of Section 3.1 instead of in Section 4, where we now state that the charging power and queue time are assumed to be the only stochastic and unknown parameters, while all others (except derived parameters) are fixed and known to the agent. We also further specify, in Sections 3.3.1 and 3.3.2 how observations of queue time and charging power may be obtained by an agent through the use of sensors in the vehicle.
>
> **"Presentation for the static problem formulation also has issues: Section 3.1 merely defines the path problem in Equation 1. Charging stations and energy consumption are not relevant here. Or did you use quantities related to charging stations and associated energy consumption in the modeling of Equation 1?"**
>
> We now clarify where Eq. 1 used, when we construct the feasibility graph in Section 3.2. We consider the road graph described in Section 3.1 (including the energy consumption and travel time of each road segment, and the parameters tied to the subset of road graph vertices containing charging stations) necessary for the later description of how we construct the feasibility graph in Section 3.2.
>
> **"In the paragraph "The general resource-constrained shortest path problem (Joksch, 1966) (with energy as the resource) is still computationally hard, especially when resource ...." Could you please specify the exact assumptions when you refer "with these assumptions", to consider computational efficiency? Without a definition of "resource-constrained shortest path problem", the statement of this graph is unclear."**
>
> We have rewritten and expanded the beginning of Section 3.2 to more clearly define how the resource-constrained shortest path problem relates to our work. In particular, we now clarify that the single-resource version of RCSPP is a special case of our problem (with consumed energy as the resource and battery capacity as the constraint), where it is also possible to replenish (recharge) the resource at a subset of the vertices, making it necessary to approximate the problem since single-resource RCSPP is NP-complete. We also clarify which assumptions we utilize when we construct the feasibility graph.
>
> Regarding the second major issue, the Bayesian approach:
>
> **"It is unclear why a Bayesian approach was taken to solve this problem, whereas the frequency-based approach is a standard solution in CMAB literature. In particular, [Chen 2013] proposed the CUCB algorithm that can be used with approximate oracles, and subsequent studies propose a variant of CUCB for extensions. HOWEVER, unless I am missing something, this study does not apply CUCB to solve their problem, and it is unclear why."**
>
> Even when both the parameter value and the prior distribution are unknown, we may, in practice, utilize partial information available (e.g., occupancy information for some charging stations) to assign prior beliefs for the parameters using this approach (enabling more efficient exploration of the charging stations than if no prior information is available, using Bayesian bandit methods like Thompson Sampling). Nonetheless, in principle, CUCB could also be applied to this problem, perhaps with an arm-specific upper confidence bound derived for sub-exponential reward distributions (Jia et al. 2021). However, in practice, the way we adapt BayesUCB to the CMAB setting is quite similar to CUCB, with base arm specific lower confidence bounds, so we expect the performance to be similar.
>
> Regarding the minor issues:
>
> **1:** All parameters are non-negative real numbers, and we now clarify this for the stochastic and unknown parameters.
>
> **2-4:** We now clarify that everything is fixed and known to the agent except the charging power (consequently: the charging time) and the queue time.
>
> **5:** We have added a comment about the assumption that each charging session has to fully charge the battery in Section 3.2, after Eq. 3.
>
> **6:** Since we believe that the road graph introduction is necessary for the description of how we construct the feasibility graph, we prefer being more explicit (and it is also referred to later, in Section 6). However, we now include an Appendix with a notation table, to increase readability.

---

### Comment · Action_Editors · 2023-09-20
**discussion period started three days ago**

Dear authors and reviewers,

I just wanted to let you know that the discussion period started four days ago, and will be ongoing for 2 weeks total. Please make sure to read all reviews and discuss any points that you disagree with (this goes for both authors and reviewers).

Best,
the action editor

---

### Author Response · Authors · 2023-11-17
**Camera Ready Revision**

We want to thank the reviewers and the action editor for their valuable constructive feedback and help in improving our manuscript! The camera ready version is now uploaded, along with a link to the open-source repository containing the code of our simulation framework.

---

### Decision · Action_Editor_38kW · 2023-11-02

**Recommendation:** Accept as is

**Comment:**

All three reviewers are in favor of acceptance, although 1 more so (accept) than the others (2 leaning accept). I concur with the reviewers that, overall, the paper deserves acceptance. As several reviewers point out, the technical novelty is slightly on the weak side. At the same time, the application and experiments are quite impressive, which outweighs the negatives.

**Audience:**

I think so. Specifically, the application seems to be of interest, and the realistic experiments are likely of interest as well. The authors promise in their response to open-source the experiments, which will hopefully be honored.

**Claims And Evidence:**

Yes, experimentally. The authors have a convincing experimental setup using road network data from three Scandinavian countries.

Theoretically, the paper is somewhat lacking. The authors give a proposed MAB approach to their problem, but no regret analysis is provided. A potential path is suggested by the authors in their rebuttal, but this argument would require some work to verify properly.